# Longitudinal Changes in the Fatty Acid Profile in Patients with Head and Neck Cancer: Associations with Treatment and Inflammatory Response

**DOI:** 10.3390/cancers14153696

**Published:** 2022-07-29

**Authors:** Constantina N Christou, Ylva Tiblom Ehrsson, Johan Westerbergh, Ulf Risérus, Göran Laurell

**Affiliations:** 1Department of Surgical Sciences, Uppsala University, 751 85 Uppsala, Sweden; ylva.tiblom.ehrsson@surgsci.uu.se (Y.T.E.); goran.laurell@surgsci.uu.se (G.L.); 2Uppsala Clinical Research Center, Uppsala University, 751 85 Uppsala, Sweden; johan.westerbergh@ucr.uu.se; 3Department of Public Health and Caring Sciences, Uppsala University, 752 37 Uppsala, Sweden; ulf.riserus@pubcare.uu.se

**Keywords:** head and neck cancer, fatty acids, inflammation

## Abstract

**Simple Summary:**

Cancer-associated malnutrition affects nutrient metabolism, including the metabolism of lipids. Toxicities associated with the treatment for head and neck cancer (HNC) may contribute to malnutrition through impaired oral intake and inflammation. Studies on lipid metabolism in patients with HNC are very limited. The anti-inflammatory effect of some fatty acids (FAs) is already proven in other cancers but the results of these studies in HNC are not consistent. This prospective study of 174 patients with HNC contributes to our knowledge of alterations in lipid metabolism following treatment for HNC and serves as basis for future research.

**Abstract:**

Studies on fatty acids (FAs) in patients with head and neck cancer (HNC) are limited. We aimed to investigate the longitudinal changes of circulating FAs in patients with HNC and to examine potential correlations of FA changes with treatment. The secondary aims were to investigate correlations of FAs with cytokines and patient-related factors, and if any FAs correlated with disease recurrence or death. A total of 174 patients with HNC were included before treatment and followed-up at three time points after the start of the treatment through blood sampling and body weight measurements. Serum FA profiling was assessed by gas chromatography. The total follow-up time was 3 years. The levels of almost all FAs changed from baseline to 7 weeks. The change in FA 14:0 was associated with treatment and the change in 18:3n-6 was associated with the patients’ pre-treatment BMI. FAs 14:0 and 18:0 were correlated with weight changes from baseline to 7 weeks. IL-6 was correlated with three FAs at 7 weeks and with two FAs at 1 year. Patients with higher levels 20:5n-3 at 3 months had a higher risk of all-cause death within 3 years (HR 2.75, 95% CI 1.22–6.21). Treatment, inflammation, and weight loss contributed in a complex manner to the altered FA profile in the studied cohort. The association between IL-6 and FAs in patients with HNC is in line with earlier studies and suggests the opportunity for regulating inflammation in HNC patients through modulation of FAs.

## 1. Introduction

An important focus of cancer research is to recognize individual condition-related factors that may affect the outcome of the treatment. One important factor influencing the outcome of treatment is the nutritional status of the patient [1]. The prevalence of malnutrition in patients with cancer is high and is associated with cancer type, site, stage, and treatment [2]. Head and neck cancer (HNC) affects the upper aerodigestive tract and is often associated with odynophagia and dysphagia, and thus patients with HNC are prone to malnutrition. Because malnutrition is a key prognostic factor for treatment success, it is important to better understand the nutritional problems of patients with HNC. Research has to extend beyond simple body weight and body composition measurements and needs to include other nutritional assessments [3].

The treatment for HNC is associated with various toxicities during the treatment and in the longer perspective after the treatment, and these may further affect oral food intake [4,5]. Although modern radiotherapy techniques are associated with fewer side effects, the risk for malnutrition is still high during and after the treatment [6]. The impaired oral intake seen in patients with HNC often leads to various nutrient deficiencies. In addition, the ability to utilize nutrients may be altered due to different metabolic changes [7]. Activation of systemic inflammation is the hallmark of cancer-associated malnutrition that can cause anorexia and a catabolic state contributing to muscle wasting and loss of adipose tissue [8,9].

Lipids, more specifically fatty acids (FAs), are the major units of energy storage and are also used as structural components in biological membranes. Lipids also have an essential role in cell signaling pathways, i.e., functioning as second messengers and as hormones. FAs can be divided into two groups, saturated FAs and unsaturated FAs. A type of unsaturated FA is the polyunsaturated FAs (PUFAs), such as omega-3 FAs (n-3 PUFAs) and the omega-6 FAs (n-6 PUFAs). Two of these FAs, 18:2n-6 (linoleic acid) and 18:3n-3 (linolenic acid), are also called essential FAs because they are obtained only through the diet. N-3 PUFAs are reported to have anti-inflammatory effects by decreasing the production of the pro-inflammatory cytokines TNF-α, interleukin-1β (IL-1β), and interleukin-6 (IL-6) and by increasing the concentration of the anti-inflammatory cytokine interleukin-10 (IL-10) [10]. The role of IL-6 in the progression of cachexia in patients with cancer was shown previously [11].

Previous studies have suggested that the administration of n-3 FAs may reverse cachexia in advanced pancreatic cancer and even modulate various inflammatory mediators such as IL-6 [12,13]. The results from studies on the effect of nutritional supplementation of n-3 on weight in patients with HNC are not consistent, possibly because of the small number of patients studied [14,15]. A study on the FA profile in patients with HNC might identify knowledge gaps and elucidate the effect of treatment on the longitudinal changes in FAs. Such data could increase knowledge on the time sequence of FA changes and could serve as background for the design of future randomized controlled studies on nutritional supplement with n-3 FAs or other potentially beneficial FAs.

This prospective study aimed to describe the longitudinal changes of circulating FAs in a large cohort of patients with HNC up to 1 year after termination of treatment and to correlate potential FA changes with treatment. The secondary aims were to assess the correlation of FA changes with gender, tumor stage, tumor site, body mass index (BMI), body weight changes, and cytokines, as well as to identify if any FAs might be related to tumor recurrence as well as all-cause death during a follow-up time up to 3 years.

## 2. Materials and Methods

This cohort, generated from a prospective observational study, included 174 patients with HNC and was carried out at three tertiary referral hospitals in Sweden. Inclusion criteria were a performance status of 0–2 according to the World Health Organization (WHO)/ECOG Performance status and curable treatment intention for a newly diagnosed and untreated HNC. Exclusion criteria were previous treatment for cancer in the last 5 years (except for skin cancer), severe alcohol problems, cognitive impairment, or inability to understand the Swedish language.

The median age of the patients was 64 years (range 34–89 years) with a male-to-female ratio of 2.8:1 (128 males, 46 females). Tumor in the oropharynx was the most common site (*n =* 78, 44.8%) followed by the oral cavity (*n =* 52, 29.9%). Most patients had stage I cancer (*n =* 74, 42.5%) according to the 8th edition of the AJCC/UICC staging system. The characteristics of the patients are described in Table 1. The majority of the patients were overweight/obese at baseline (*n =* 105, 60%) with a BMI ≥ 25 or BMI ≥ 27 if ≥ 70 years.

The patients were included before treatment at the tertiary referral hospital and were followed up at 7 weeks after the start of the treatment and at 3 months and 1 year after the termination of treatment. The patients were assessed by blood sampling and body weight measurements at the previously mentioned regular intervals. The follow-up time was 2 years for recurrence and disease-specific death and 3 years for all-cause death. The follow-up was, for some patients, carried out at the local hospital due to long travel distance to the tertiary referral hospital. There were some missing values at each time point because of death and of patients not being able to show up for the planned visit to the hospital, mostly because of fatigue. The outcomes of 27 out of the 174 patients have been reported earlier in an explorative study [16].

The blood samples were analyzed for FAs and cytokines after storage at –70 °C at the Uppsala Biobank, Sweden. FA profiling was assessed by gas chromatography. For the analyses, 50 μL of centrifuged clarified serum (2000× *g* for 10 min) was used as the sample. After extracting serum overnight with a methanol/chloroform/butylated hydroxytuluene/NaH_2_PO_4_ solution, cholesteryl esters were separated by TLC before subsequently being transmethylated and separated by GLC on a 30-m glass capillary column coated with Thermo TR-FAME (Thermo Fisher Scientific, Waltham, MA, USA) with helium as a carrier gas. The system used included a model GLC 6890 N, autosampler 7683, and Agilent ChemStation (all from Agilent Technologies, Waldbronn, Germany). The temperature was programmed to 150–260 °C. Individual FAs were detected by flame ionization detection and identified by comparing retention times to FA methyl ester standards Nu Check Prep [17]. FA profile included unsaturated FAs 16:1n-7 (palmitoleic acid) and 18:1n-9 (oleic acid), PUFAS 18:2n-6 (linoleic acid), 18:3n-6 (gamma-linolenic acid), 18:3n-3 (linolenic acid), 20:3n-6 (dihomo gamma-linolenic acid), 20:4n-6 (arachidonic acid), 20:5n-3 (eicosapentaenoic acid), 22:6n-3 (docosahexaenoic acid), saturated FAs 14:0 (myristic acid), 15:0 (pentadecanoic acid), 16:0 (palmitic acid), and 18:0 (stearic acid). The values of the FAs are presented as percentages of the total amount of FAs. Serum was also analyzed for cytokines IL-1alpha, IL-2, IL-6, and IL-10, among other protein biomarkers, using an immuno-oncology biomarker panel (OLINK, Uppsala, Sweden). The panel offered simultaneous multiplex immunoassay analysis of 92 protein biomarkers, including the cytokines evaluated in the present study. The results are presented as normalized protein expression, an arbitrary unit in Log2 scale. The cytokine serum levels have been reported earlier [18].

Body weight (kg) measurements were performed in patients without outdoor clothing and shoes at follow-ups using a weight scale. Height was measured by a research nurse using a stadiometer with the patient in the Frankfort plan. BMI was calculated using the formula kg/m^2^. The patients were divided into underweight, normal weight, overweight, and obese based on their BMI and according to Global Leadership Initiative on Malnutrition (GLIM) [19].

Patients were under nutritional surveillance according to hospital guidelines, and nutritional support was offered when indicated. Nutritional treatment included enteral nutrition (total or partial) through either a nasogastric tube or gastrostomy tube and parenteral nutrition (total or partial). BMI and nutritional parameters of the patients are described in Table 2. Patients were stratified into four treatment groups: radiotherapy ± surgery, chemo radiotherapy ± surgery, radiotherapy and cetuximab ± surgery, and surgery only.

### Statistical Analysis

The FA changes over time were analyzed by plotting the FA values together with the mean value at each time point. The effects were tested for differences between time points by fitting a linear regression model with patient ID as the random effect and time point as a fixed effect. For the analysis of the change in FA levels over time, a comparison was made between each time point versus the pre-treatment point.

The association between change of FAs over time and the individual patient-related factors of interest (treatment, gender, tumor stage, tumor site, and BMI pre-treatment) was examined by a multivariable linear regression model, creating five models that in addition to patient ID and time points also included all five main effects and the interaction between one of the main effects and time point. BMI was modeled as a continuous variable but plotted in its four BMI categories (underweight, normal weight, overweight, and obese).

The association between cytokine levels and FAs was first examined by fitting a linear regression model for each cytokine and time point, with cytokine as a dependent variable and all FAs as independent variables. For cytokines that showed an association, the cytokine level was plotted for each FA and time point and the Spearman correlation was calculated and tested.

The body weight changes in percent in relation to the pre-treatment body weight were plotted against FA for each time point and analyzed with the Spearman correlation test.

The FA levels at 3 months for individuals that experienced a relapse within 2 years from the end of the treatment were compared with the FA levels of patients with loco-regional control. The levels were plotted and the differences were tested using a *t*-test. The ratio 16:1n-7/16:0 was plotted and tested using a *t*-test for the outcomes of recurrent disease, all-cause death, and disease-specific death. Cox proportional hazard models were fitted for the previously mentioned outcomes.

Levels of 18:3n-3, 20:5n-3, and 22:6n-3 at 3 months were compared between patients who died and patients who survived at 3 years. Cox proportional hazard models were fitted using all-cause death as the outcome.

Due to multiple comparisons, the *p*-values were adjusted using the Holm–Bonferroni method [20]. A *p*-adjusted value (*p*-adj) ≤ 0.05 was considered statistically significant. All analyses were performed in R version 3.6.1 (5 July 2019).

## 3. Results

### 3.1. FA Changes over Time

All FAs showed a significant change over time, although the direction and size of the change varied considerably. The levels of all four saturated FAs changed in a significant way between the pre-treatment time (baseline) and at 7 weeks after the start of treatment. FA 14:0 (*p*-adj < 0.001), 15:0 (*p*-adj = 0.023), and 18:0 (*p*-adj < 0.001) decreased between these two time points, and 16:0 increased (*p*-adj = 0.007) (Figure 1). However, the clearest pattern was for 14:0, which showed a nadir during the treatment at 7 weeks, and thereafter a linear and time-dependent increase from 7 weeks until 1 year. The values of the above-mentioned FAs showed a change from 7 weeks to 1 year and then returned to baseline levels, as shown in Figure 1.

Most unsaturated FAs showed a significant change between baseline and 7 weeks. FAs 16:1n-7 (*p*-adj < 0.001) and 18:3n-6 (*p*-adj < 0.001) decreased between these time points, while 20:4n-6 (*p*-adj < 0.001), 20:3n-6 (*p*-adj = 0.018), and 22:6n-3 (*p*-adj < 0.001) increased (Figure 1). The levels of these unsaturated FAs changed from 7 weeks to 1 year reaching almost the pre-treatment levels. For 18:2n-6 the 1-year level was higher than baseline (*p*-adj = 0.023), and for 20:3n-6 the 1-year level was lower than the baseline level (*p*-adj < 0.001) (Figure 1).

### 3.2. Correlation between FAs and Treatment and Other Patient-Related Factors

From the multivariable linear regression test, only 14:0 levels were significantly associated with treatment modalities (*p*-adj = 0.023). The surgery group showed a stable course of 14:0 levels over time, while the other three treatment groups showed an initial decrease of 14:0 followed by a recovery to the pretreatment levels, as shown in Figure 2.

An association was seen between the level of 18:3n-6 and BMI before treatment (*p*-adj = 0.045). The pattern of change in the levels of 18:3n-6 for the overweight and obese groups displayed a decrease from baseline to 7 weeks and an increase from 7 weeks to 1 year after the end of treatment. The normal weight group showed a stable pattern, and the underweight group showed a slight increase from baseline to 3 months (Figure 3).

No association was observed between the FA profile and gender, tumor stage, or tumor site.

### 3.3. Correlation of FAs with Body Weight Changes

Spearman correlation tests showed a correlation between body weight change during the treatment period (between baseline and 7 weeks) and four FAs. Changes in 14:0 (r_s_ = 0.466, *p*-adj < 0.001) and 18:0 (r_s_ = 0.323, *p*-adj = 0.001) showed a positive correlation with body weight change, whereas changes in 20:4n-6 (r_s_ = −0.374, *p*-adj < 0.001) and 22:6n-3 (r_s_ = −0.436, *p*-adj < 0.001) showed a negative correlation. This is demonstrated in Figure 4.

There was no association between body weight change from baseline to 3 months or from baseline to 1 year after the termination of treatment and any of the FAs.

### 3.4. Correlations between FAs and Cytokines

From the linear regression analysis, only IL-6 among the four tested cytokines showed an association with FAs at 7 weeks (*p*-adj = 0.003) and at 1 year from the termination of treatment (*p*-adj = 0.002). Spearman correlation showed a negative correlation between IL-6 and 14:0 at 7 weeks (r_s_ = −0.265, *p*-adj = 0.038), a negative correlation between IL-6 and 18:0 at 7 weeks (r_s_ = −0.292, *p*-adj = 0.01), and a positive correlation between IL-6 and 20:3n-6 at 7 weeks (r_s_ = 0.268, *p*-adj = 0.033). The correlations of all FAs with IL-6 at 7 weeks are demonstrated in Figure 5.

At 1 year, IL-6 showed a positive correlation with 16:0 (r_s_ = 0.278, *p*-adj = 0.033) and a negative correlation with 18:2n-6 (r_s_ = −0.272, *p*-adj = 0.042), as shown in Figure 6.

### 3.5. FAs as Prognostic Factors for Recurrence and Mortality

Within 2 years from the end of treatment, 50 patients out of 174 patients (29%) experienced disease recurrence and 33 patients (19%) were deceased because of their cancer. All-cause death within 3 years was observed in 38 patients (22%). It was anticipated that the levels of FAs at 3 months after treatment were most predictive as prognostic factors.

The levels of FAs at 3 months after the end of treatment were not associated with disease recurrence within 2 years. Moreover, the pretreatment 16:1/16:0 ratio did not predict overall outcomes, i.e., disease recurrence, disease-specific death, or all-cause death within 2 or 3 years from the end of treatment in the study population.

However, all-cause death was associated with 20:5n-3 (*p*-adj = 0.044). Patients with higher levels of 20:5n-3 at 3 months after the end of treatment had a higher risk of all-cause death within 3 years (HR 2.75, CI 1.22–6.21), as demonstrated in Figure 7.

## 4. Discussion

Circulating FAs were studied in a mixed population of patients with HNC from pre-treatment to 1 year after the termination of treatment. It was demonstrated that almost all studied FAs changed in a significant way between pre-treatment and 7 weeks, which was the treatment duration for most patients. Between these two time points the saturated 14:0, 15:0, 18:0, the monounsaturated 16:1n-7, and the PUFA 18:3n-6 decreased, while saturated 16:0 and the three PUFAs 20:4n-6, 20:3n-6, and 22:6n-3 increased.

Longitudinal changes of circulating FAs in HNC patients using repeated assessments have rarely been studied previously. The observations of early FA changes in the current study provide strong evidence that most alterations were attributed to treatment-induced body weight changes and systemic inflammation. HNC treatment is known to be associated with weight loss in the majority of patients, and recent publications have described that treatment modalities such as radiotherapy and chemo radiotherapy induce a systemic inflammation response [21]. In the present study, IL-6 was used as a systemic inflammation response indicator. For patients undergoing surgery alone, the level of 14:0 remained stable through the study period, which is in line with the assumption that surgery is less likely to cause systemic effects. However, the molecular events leading to alterations of circulating FAs in connection to treatment remain elusive.

Only a few previous studies have reported levels of FAs after cancer treatment. A study on 42 men with squamous cell esophageal cancer observed higher levels of 22:6n-3 and 20:4n-6 after chemo radiotherapy, which is in agreement with the findings of the present study [22]. However, the higher levels of the above-mentioned FAs could not be correlated to treatment in the present cohort. An improved FA and lipid profile with an increase of PUFA and a decrease in low-density lipoproteins was observed after radiotherapy in patients with breast cancer in comparison with healthy controls [23].

Although the FA profile was unaffected by cancer site and stage, changes in the underlying metabolic mechanism due to cancer site-specific pathology may still have been present. However, the fact that the proportion of patients in need of nutritional support was highest (26.4%) at 7 weeks further strengthens that the changes in the FA profile at this time point were related to treatment-induced impaired energy intake and subsequent weight loss. Indeed, correlations of four of the studied FAs with body weight change were observed between pretreatment and 7 weeks. The treatment period is associated in many patients with dysphagia and increased malnutrition rates [6,24]. Multiple physiological mechanisms can thus have contributed to the observed increased levels of both 22:6n-3 and 20:4n-6 in association with body weight loss. Alteration of the circulating FAs, including the previously mentioned FAs, was reported in patients 1 year after undergoing bariatric surgery where the observed changes in serum FA profile were associated with alterations in endogenous lipid metabolism due to weight loss [25].

Circulating levels of the n-3 PUFA 18:3n-3, an essential FA, is less reflective of its oral intake in comparison to the other n-3 PUFAs because 18:3n-3 serves as precursor metabolite from which n-3 PUFAs such as 20:5n-3 and 22:6n-3 are synthesized via elongation and desaturation [26]. It seems as though 18:3n-3 is more easily oxidized during the postprandial period than other PUFAs [27], although data are limited and this result has not been confirmed in a fasting state [28]. The other essential FA 18:2n-6 is the precursor from which the n-6 PUFAs are synthesized, and it is a good marker of dietary 18:2n-6 intake from, for example, vegetable oils. Values of two n-6 PUFAs at 1 year differed from pre-treatment values. FA 18:2n-6 was higher and the value of 20:3n-6 was lower. This pattern has been associated with an improved metabolic profile and has been linked to lower cardiometabolic disease risk and might reflect either altered delta-6 and delta-5 desaturase activities during the follow-up or dietary changes [29]. These two n-6 PUFAs are dependent on each other because 20:3n-6 is elongated and desaturated from 18:2n-6 and further desaturated to n-6 PUFAs such as 20:4n-6 [30]. The lower level of 20:3n-6 at 1 year indicates alterations in lipid metabolism in patients with HNC even 1 year after treatment termination [16].

An increased level of systemic inflammation is reported to affect patients undergoing treatment for HNC. There are a number of findings showing that the pathogenesis of inflammation is multifactorial and cannot be related only to the cancer disease. Many inflammation-based scores such as the ratio of C-reactive protein to albumin are related to prognosis of several cancers, including HNC [31]. Radiotherapy and chemo radiotherapy induce an immune response including the release of pro-inflammatory cytokines, which contributes to the increased level of systemic low-grade inflammation [32]. Pro-inflammatory cytokines such as IL-6 have been related to oral mucositis induced by treatment [33]. In addition, obesity is reported to independently contribute to low-grade inflammation through the production of pro-inflammatory cytokines [34]. There are grounds for obesity’s effects on systemic inflammation in the present cohort as 60 percent of the patients were overweight or obese before treatment.

In the present study, the pro-inflammatory cytokine IL-6 showed a negative correlation with 14:0 and 18:0 and a positive correlation with 20:3n-6 at 7 weeks. IL-6 cytokines are involved in inflammation by controlling the differentiation, proliferation, migration, and apoptosis of immune cells. Additionally, IL-6 plays a role in a variety of other processes such as metabolism, but its complete role in cancer immunology is not fully characterized. High levels of circulating IL-6 have, in several studies, been correlated with poor survival and oncological outcomes in patients with HNC [35].

Il-6 is secreted mainly by macrophages, but also from adipose tissue, and is known to have a vital role in lipid metabolism [36]. The complex interplay is further reflected by the influence of FAs on inflammation parameters. Saturated FAs have been linked to inflammation promotion, while PUFAS and monounsaturated FAs have been mostly shown to induce anti-inflammatory effects [37]. The role of saturated FAs in inflammation has been studied in vitro, and the production of IL-6 increased when human skeletal muscle cells were stimulated with 16:0 [38]. In contrast, FA 18:2n-6 prevented the 16:0-induced upregulation of IL-6 and thus seems to have an immunosuppressive action [38].

Inflammation observed in the tumor microenvironment (TME) has many tumor-promoting effects such as proliferation and survival of malignant cells, promotion of angiogenesis and metastasis, inhibition of immune responses, and altered responses to hormones and chemotherapeutic agents [39]. Steady interest in the TME has led to numerous publications discussing the importance of the altered lipid metabolism in the TME that mediates immunosuppressive mediators [40]. Therefore, a possible target of nutritional support could be the reduction of the immunosuppressive mechanisms in the TME in addition to the suppression of the systemic inflammatory state of the patients [41,42]. The reduction of several inflammatory markers due to administration of n-3 PUFAs was shown in a meta-analysis of colorectal cancer and a meta-analysis of gastric cancer [43,44]. Solis-Martinez et al. [45] observed a decrease in serum levels of pro-inflammatory cytokines and regulation of body weight when supplementing the n-3 FA 20:5n-3 during treatment in patients with HNC. A study in 31 patients with stage III and IV HNC suggested an improved inflammatory state when receiving oral supplementation containing n-3 PUFAs during chemo radiotherapy although the results need confirmation through larger randomized, controlled trials [46].

Different FAs influence not only the inflammatory state, but also health in general. The beneficial role of 20:5n-3 and other n-3 PUFAs in cardiovascular health has been suggested, and a recent pooled analysis of 17 prospective cohort studies reported that high circulating levels of 22:6n-3 and 20:5n-3 have positive health effects and lower risk of premature death [47]. In the present study, deaths from any cause were included as the outcome. An unexpected finding was the increased risk of all-cause death within 3 years in case of higher levels of 20:5n-3 at 3 months after the treatment termination, but this was a secondary analysis and thus needs confirmation to exclude a chance finding. The role of 20:5n-3 in HNC is mostly studied regarding its role in body weight changes and inflammation markers, with inconsistent results suggesting the need for further studies [45,48].

The present study has some limitations. First, there was no control group that could have been used to compare the pre-treatment FA profile of the patients with HNC and thus elucidate the impact of cancer on FA profiles. Second, the mixed HNC population had different cancer sites, and a more homogeneous group with regard to tumor site would have been preferable. Third, elevated inflammation is reported to be attributed to a high-volume adipose tissue, a measure which was not investigated in the present study population.

## 5. Conclusions

In conclusion, our study suggests that treatment-induced effects such as impaired food intake, body weight loss, and systemic inflammation contribute in a complex manner to the altered FA profile seen in the current cohort. The association between IL-6 and FAs even in patients with HNC is in line with earlier studies and suggests the opportunity for regulating inflammation in HNC patients through modulation of FAs.

## Figures and Tables

**Figure 1 cancers-14-03696-f001:**
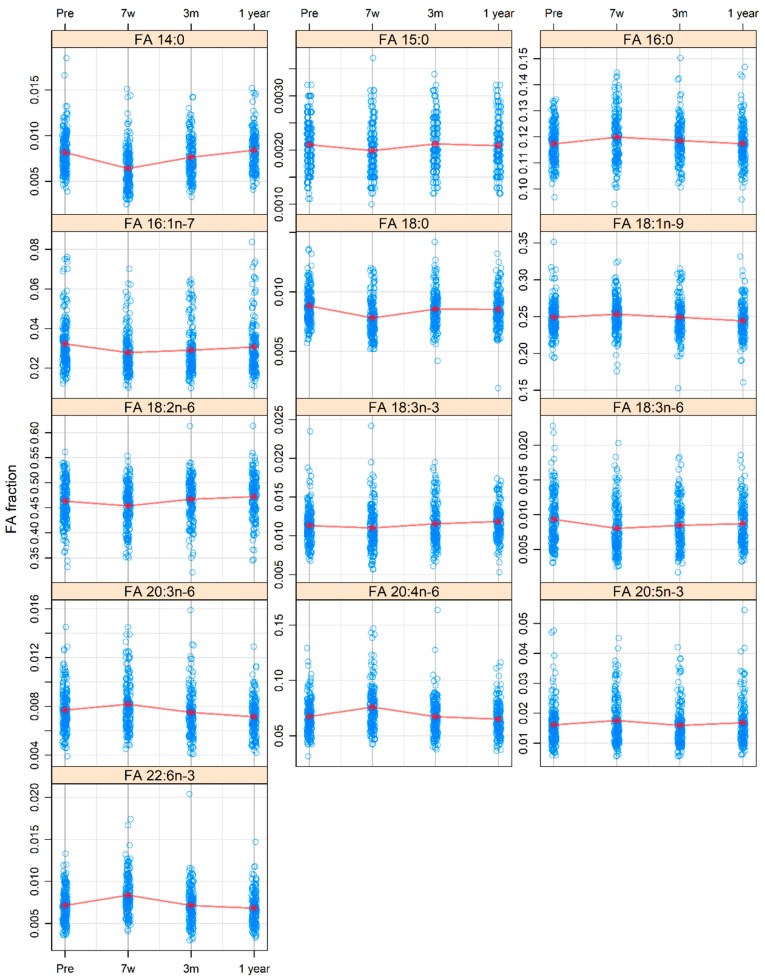
Serum levels of fatty acids (FAs) 14:0, 15:0, 16:0, 16:1n-7, 18:0, 18:1n-9, 18:2n-6, 18:3n-6, 18:3n-3, 20:3n-6, 20:4n-6, 20:5n-3, and 22:6n-3 at pre-treatment (pre), at 7 weeks after the start of treatment (7 w), and at 3 months (3 m) and 1 year (1 year) after the termination of treatment in patients with head and neck cancer. The values for the FAs are presented in relation to the total amount of FAs.

**Figure 2 cancers-14-03696-f002:**
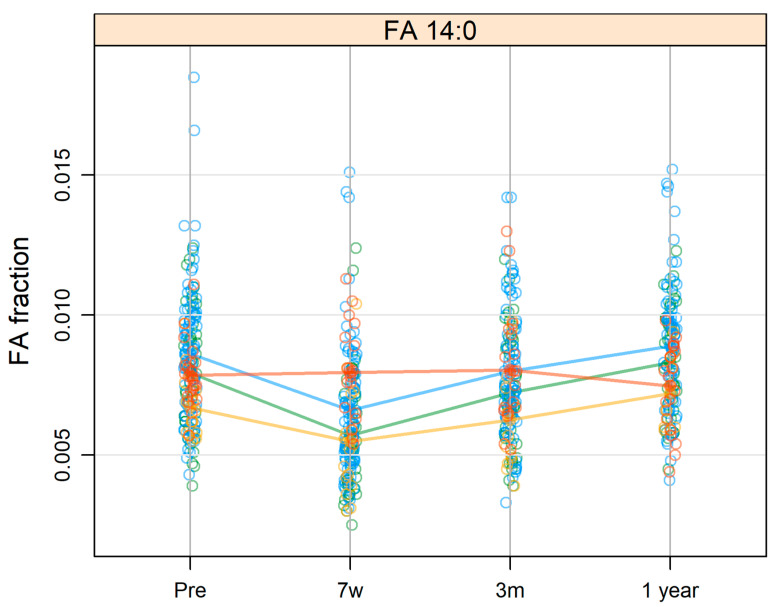
Levels of FA 14:0 in patients with head and neck cancer divided into different treatments (red: surgery; blue: radiotherapy; green: chemoradiotherapy; yellow: radiotherapy and cetuximab) at pre-treatment (pre), at 7 weeks after the start of treatment (7w), and at 3 months (3m) and 1 year (1 year) after the termination of treatment.

**Figure 3 cancers-14-03696-f003:**
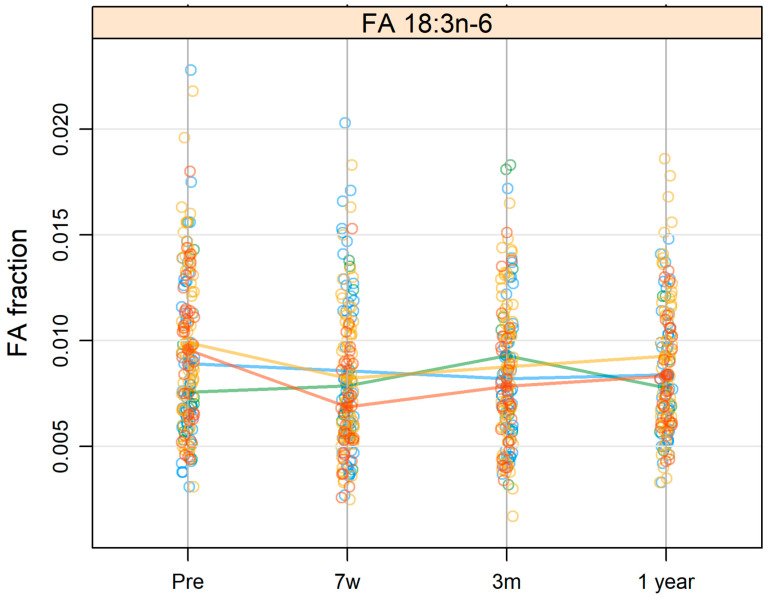
Levels of FA 18:3n-6 in patients with head and neck cancer divided into Body mass index (BMI) groups (green line: underweight;, blue line: normal weight;, yellow line: overweight;, red line: obese) at pre-treatment (pre), at 7 weeks after the start of treatment (7w), and at 3 months (3m) and 1 year (1 year) after the termination of treatment. The values of the FA are presented in relation to the total amount of FAs.

**Figure 4 cancers-14-03696-f004:**
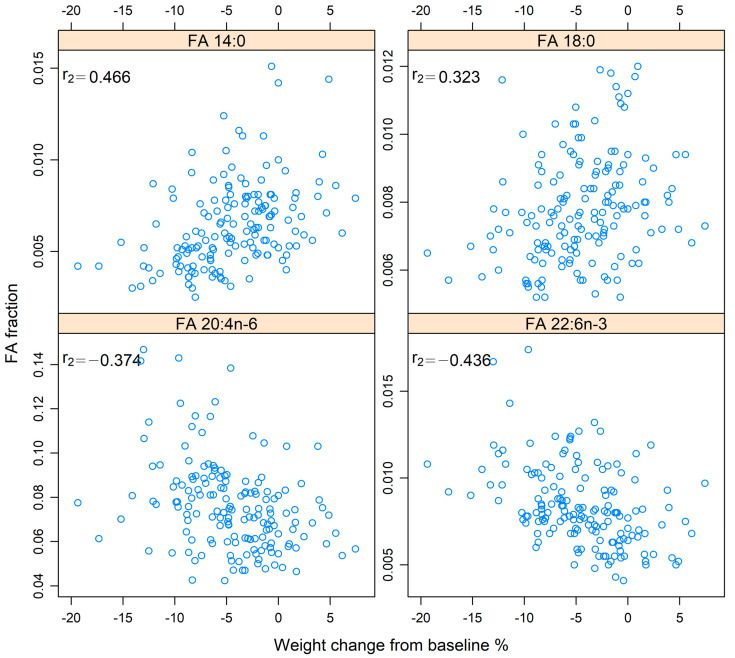
Spearman’s rank correlation (r_s_) between fatty acids (FAs) 14:0, 18:0, 20:4n-6, and 22:6n-3 and percent body weight changes from baseline to 7 weeks in patients with head and neck cancer. The values for the FAs are presented in relation to the total amount of FAs. The *x*-axis shows percent body weight change from baseline.

**Figure 5 cancers-14-03696-f005:**
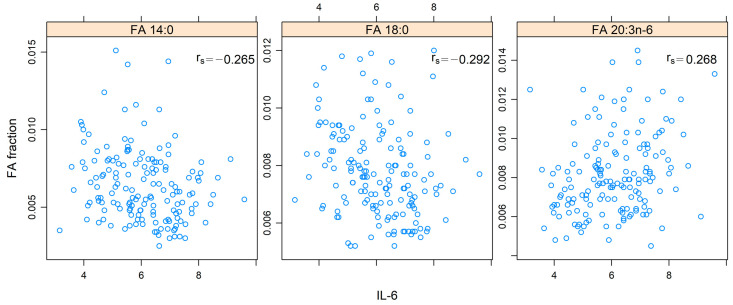
Spearman´s rank correlation (r_s_) of fatty acids (FAs) 14:0, 18:0, and 20:3n-6 and interleukin 6 (IL-6) at 7 weeks in patients with head and neck cancer. The values for the FAs are presented in relation to the total amount of FAs. The values of IL-6 are presented in log-2 scale on the *x*-axis.

**Figure 6 cancers-14-03696-f006:**
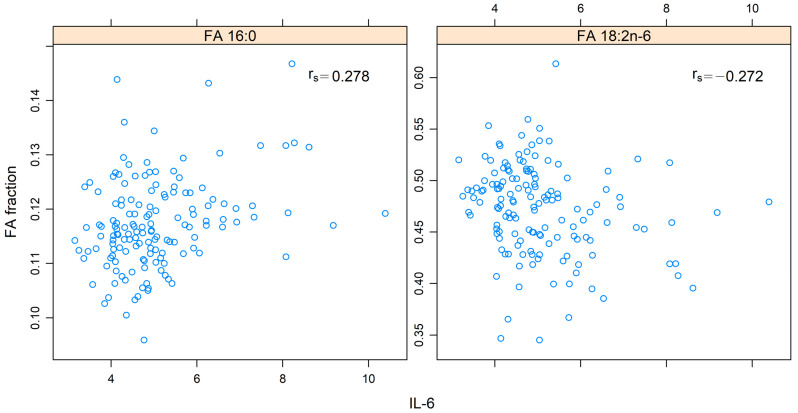
Spearman´s rank correlation (r_s_) between fatty acids (FAs) 16:0 and 18:2n-6 and interleukin 6 (IL-6) at 1 year after the end of treatment in patients with head and neck cancer. The values for the FAs are presented in relation to the total amount of FAs. The values of IL-6 are presented in log-2 scale on the *x*-axis.

**Figure 7 cancers-14-03696-f007:**
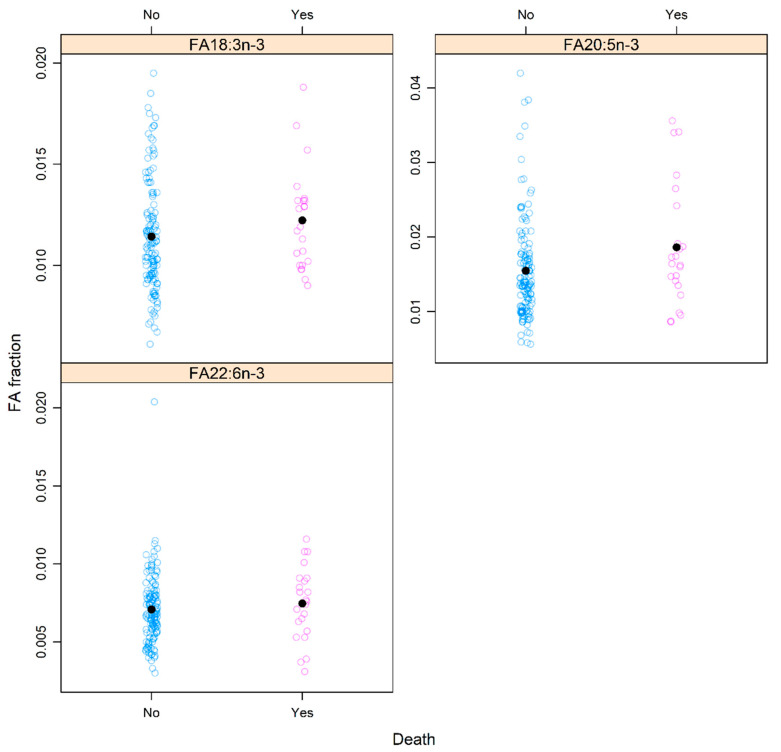
Association between all-cause death (yes = death, no = alive) and levels of fatty acids (FAs) 18:3n-3, 20:5n-3, and 22:6n-3 at 3 months after the end of treatment.

**Table 1 cancers-14-03696-t001:** Characteristics of patients with head and neck cancer (*n* = 174).

Age, Median (Q1–Q3)	64 (56–71)
Gender, *n* (%) Male Female	128 (73.6) 46 (26.4)
Tumor site, *n* (%) Oropharynx Oral cavity Larynx Hypopharynx Nasopharynx Unknown primary Other *	78 (44.8) 52 (29.9) 20 (11.5) 5 (2.9) 3 (1.7) 3 (1.7) 13 (7.5)
Tumor stage UICC version 8, *n* (%) I II III IV	74 (42.5) 34 (19.5) 28 (16.1) 38 (21.8)
Treatment type, *n* (%) Surgery Radiotherapy ± surgery Chemoradiotherapy ± surgery Radiotherapy + cetuximab ± surgery	24 (13.8) 89 (51.1) 46 (26.4) 15 (8.6)

* Cancer of the external auditory canal (*n* = 2), salivary gland cancer (*n* = 4), sinonasal cancer (*n* = 6), and outer ear cancer (*n* = 1).

**Table 2 cancers-14-03696-t002:** Body mass index (BMI) and nutritional parameters of the study cohort (*n* = 174).

Study Time Points	Pre-Treatment *N* = 174	7 Weeks *N* = 159	3 Months *N* = 153	12 Months *N* = 147
BMI * kg/m^2^ (Q1–Q3) Underweight Normal weight Overweight Obese	26.4 (23.8–29.3) 16 (9.2%) 53 (30.5%) 69 (39.7%) 36 (20.7%)	25 (22.9–27.6) 22 (13.8%) 66 (41.5%) 47 (29.6%) 24 (15.1%)	24.5 (22.5–27.5) 23 (15%) 66 (43.1%) 47 (30.7%) 17 (11.1%)	24.6 (22.4–27.6) 18 (12.2%) 73 (49.7%) 37 (25.2%) 19 (12.9%)
Nutrition Oral Enteral/parenteral **	171 (98.3%) 3 (1.7%)	117 (73.6%) 42 (26.4%)	144 (94.1%) 9 (5.9%)	139 (94.6%) 8 (5.4%)

* Underweight: BMI < 20 (BMI <  22 if  ≥70 years); normal weight: BMI = 20–24.99 (BMI = 22–26.99 if ≥70 years); overweight: BMI = 25–29.99 (BMI = 27–29.99 if  ≥70 years); obese: BMI ≥ 30. ** Partially and totally enteral/parenteral nutritional support.

## Data Availability

Data is maintained in this article. Data is not publicly available due to privacy.

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
