# Peer review of "Longitudinal Changes in the Fatty Acid Profile in Patients with Head and Neck Cancer: Associations with Treatment and Inflammatory Response"

_cancers, 2022, doi:10.3390/cancers14153696_

Round 1

Reviewer 1 Report

In the presented manuscript Authors present the findings of the study concerning alterations of fatty acids (FA) profile in relation to the treatment of head and neck cancer (HNC). In general I find this paper interesting, as the perspective of predicting anticipation of  therapy outcome on the basis of preliminary analyses (FA profile in presented case) seems to be interesting and is in line with the ‘personalized medicine” trend. However, I find some flaws in presented manuscript, which I believe will improve the general impression and value of the work when explained.

1)      The simple summary and abstract are generally acceptable and attractive, giving an impression of inviting the reader to get to know the full article. What I miss is mentioning the material of FA analyses (serum) and the method of FA analysis in the abstract.

2)      Introduction section is comprehensive, the scientific background and aims are well described.

3)      Methods section – the section I have the highest concerns about.

a)       Anthropometric measurements – I believe that Authors should provide more detailed information about how measurements were performed to ensure reproducibility of the experiment. So the information about how the height measurements were performed (essential for BMI calculations – was it with professional Stadiometer in position so called  Frankfort plane?); was the weight measurements performed before/after meal in similar time of the day?

b)      Also the description of the FA analysis is very modest. I understand that there is a reference, yet some brief description about the chromatographic analysis should be included: chromatograph model, applied detection method, were some internal standards used? How the identification of individual FA’s was performed?

c)       It is also advisable to calculate the FA’s levels not only as % of total FA’s, but also in absolute values which is more precise. This is due to the fact that the % is very sensitive to analysis. it is enough to incorrectly assign a single peak of FA and it significantly changes the % values.

4)      Presentation of the results is generally solid and I like it very much; figures are clear and legible. The idea of presenting plots is interesting however I miss the general summary table of determined FA

5)      Discussion -  the discussion is interesting, but somewhat in my impression a bit of superficial; as I mentioned, I find the idea of the study extremely interesting, but the study has its limitations; It’s hard to attribute the changes of FA only to the treatment method (surgery) as long as we don’t know the details of the patients’ nutrition pattern? (special diet, dietary supplements etc.) lines 420-431 the attribution of increased inflammation to the overweight may not entirely reflect the reality – what I mean is that authors classified patients to overweight category based on BMI (which is in line with various guidelines) yet the elevated inflammation is mainly attributed to elevated adipose tissue levels – in this paper Authors did not assessed/measured fat tissue levels so it should be mentioned in limitations.

I believe that Authors would be kind to adress these comments and will improve this very interesting work

Author Response

Dear reviewer,

We thank you for the many valid and helpful comments and suggestions. We have made changes accordingly and are now sending a revised version of the manuscript “Longitudinal changes in the fatty acid profile in patients with head and neck cancer: associations with treatment and inflammatory response.” Below is a point-by-point list of changes and comments to your suggestions.

Sincerely,

Constantina Nadia Christou

Response to reviewer 1

Comment 1: The simple summary and abstract are generally acceptable and attractive, giving an impression of inviting the reader to get to know the full article. What I miss is mentioning the material of FA analyses (serum) and the method of FA analysis in the abstract.

Response: We agree with the suggestion.

Change in the manuscript: Page 1, line 25. The missing information is added in the abstract.

Comment 2:   Introduction section is comprehensive, the scientific background and aims are well described.

Response: Thank for your kind comment

Comment 3a: Anthropometric measurements – I believe that Authors should provide more detailed information about how measurements were performed to ensure reproducibility of the experiment. So the information about how the height measurements were performed (essential for BMI calculations – was it with professional Stadiometer in position so called  Frankfort plane?); was the weight measurements performed before/after meal in similar time of the day.

Response: We agree with the suggestion

Change in the manuscript: Page 4,line 142-143. Information about weight and height measurements is added.

Comment 3b:  Also the description of the FA analysis is very modest. I understand that there is a reference, yet some brief description about the chromatographic analysis should be included: chromatograph model, applied detection method, were some internal standards used? How the identification of individual FA’s was performed?

Response: We agree that this part can be expanded despite a reference with more detailed description is included in the manuscript.

Change in the manuscript: Page 3 and 4, line 119-128. The method description is updated and more information is included that is also described in our cited reference.

Comment 3c: It is also advisable to calculate the FA’s levels not only as % of total FA’s, but also in absolute values which is more precise. This is due to the fact that the % is very sensitive to analysis. it is enough to incorrectly assign a single peak of FA and it significantly changes the % values.

Response: Thank you for this good comment. Unfortunately, we only have the relative values of the fatty acids, which could be a slight limitation as the reviewer suggests. However, we wish to notify the reviewer that presenting the proportions of fatty acids is the most conventional way to present fatty acid data and there are also advantages to this approach as compared to presenting the data using absolute values (REF: Bradbury KE et al. Serum fatty acids myristic acid and linoleic acid are better predictors of serum cholesterol concentrations when measured as molecular percentages rather than as absolute concentrations. Am J Clin Nutr 2010 Feb;91(2):398-405).

For example, for myristic acid (which was a fatty acid of special interest for the results in the current study), it has been shown that when it is measured as molecular percentages, but not as concentrations, predicts serum total cholesterol levels in a manner that distinguishes between the differential cholesterolemic effects of dietary saturated and polyunsaturated fats. Thus, in that context, the percentage values were more useful and sensitive than the absolute concentrations, where the latter values might be to a larger extent influenced by the total levels of the blood lipid fractions, partly independent of diet.

No change in the manuscript.

Comment 4: Presentation of the results is generally solid and I like it very much; figures are clear and legible. The idea of presenting plots is interesting however I miss the general summary table of determined FA

Response: The names of the FAs included in the study are mentioned under the Material and Methods section, lines 128-133. Therefore, we chose not to present the FAs in a table. No change in the manuscript.

Comment 5:   Discussion -  the discussion is interesting, but somewhat in my impression a bit of superficial; as I mentioned, I find the idea of the study extremely interesting, but the study has its limitations; It’s hard to attribute the changes of FA only to the treatment method (surgery) as long as we don’t know the details of the patients’ nutrition pattern? (special diet, dietary supplements etc.) lines 420-431 the attribution of increased inflammation to the overweight may not entirely reflect the reality – what I mean is that authors classified patients to overweight category based on BMI (which is in line with various guidelines) yet the elevated inflammation is mainly attributed to elevated adipose tissue levels – in this paper Authors did not assessed/measured fat tissue levels so it should be mentioned in limitations.

Response: Thank you for these interesting comments. We agree that more details about the patients ´ nutrition patterns could help us to better understand other factors that contributed to the altered FA profile of the patients. Unfortunately, we don’t have that detailed information on the content of the patients´nutrition patterns besides the information given in table 2, page 4. One should remember that it would be hard to assess the nutrition pattern of each patient up to one year after the end of treatment. E.g. the degree of compliance in out-patients receiving enteral nutrition would be difficult to investigate.

We agree that elevated inflammation could most probably be attributed to high-volume adipose tissue but we haven´t assessed adipose tissue volume in our study population.

Change in the manuscript: Page 13, lines 405-407 A sentence about the limitation of not measuring adipose tissue volume is added.

Reviewer 2 Report

In this study, the authors investigated the longitudinal changes of circulating fatty acids (FAs) in patients with head and neck cancer (HNC) and the potential correlations of FA changes with treatment, cytokines and BMI.

I have several concerns:

1.      The authors detected the levels of serum cytokines, only including IL-1 alpha, IL-2, IL-6, and IL-10. However, other cytokines, such as TNF-α, IL-1β and TGF-β and so on, are also worth investigating.

2.      Different treatment methods and different tumor types may influence the longitudinal changes and the potential correlations of FA changes with treatment, which were not well explored in the study.

3.      The authors analyzed the prognostic value of the levels of FAs at 3 months after treatment in recurrence and mortality. Actually, the baseline levels of FAs were usually paid more attention to.

4.      It seems that in the linear regression analysis, correlations between FAs and cytokines were not adjusted for age, sex, treatment, tumor types and other covariates, which were not rigorous.

Author Response

Dear reviewer,

We thank you for the many valid and helpful comments and suggestions. We have made changes accordingly and are now sending a revised version of the manuscript “Longitudinal changes in the fatty acid profile in patients with head and neck cancer: associations with treatment and inflammatory response.” Below is a point-by-point list of changes and comments to your suggestions.

Sincerely,

Constantina Nadia Christou

Response to reviewer 2

Comment 1:  The authors detected the levels of serum cytokines, only including IL-1 alpha, IL-2, IL-6, and IL-10. However, other cytokines, such as TNF-α, IL-1β and TGF-β and so on, are also worth investigating.

Respons: We agree that other cytokines are also worth investigating. We chose to include only the above 4 cytokines because they represent both anti-inflammatory and pro-inflammatory cytokines. Including more cytokines could havedone the statistical analyses more difficult to interpret because of multiple testing. No change in the manuscript.

Comment 2: Different treatment methods and different tumor types may influence the longitudinal changes and the potential correlations of FA changes with treatment, which were not well explored in the study.

Respons: In our study we included patients with different treatments as well as different tumor sites as you can see in table 1. The majority of the patients had squamous cell carcinoma. The association between the change of FAs and different treatments as well as different tumor sites was examined by a multivariable linear regression model, please see page 4, line 163. The results of the analysis can be found in page 7-8, lines 224-245. No change in the manuscript.

Comment 3: The authors analyzed the prognostic value of the levels of FAs at 3 months after treatment in recurrence and mortality. Actually, the baseline levels of FAs were usually paid more attention to.

Respons: Thank you for your remark. We chose to analyze the prognostic value of the levels of FAs at 3 months as we consider the patients being cancer free at 3 months and the levels of the FAs not influenced by cancer as they would have been at baseline, before treatment. No change in the manuscript.

Comment 4:  It seems that in the linear regression analysis, correlations between FAs and cytokines were not adjusted for age, sex, treatment, tumor types and other covariates, which were not rigorous.

Respons: Thank you for your remark. The purpose of the linear regression was to get an overall test of the association between FAs and cytokines before we started looking at the correlation between individual FAs and cytokines . Adjusting for other variables in the linear regression would give a different overall test than the individual correlations since the adjusted linear regression would show the association between FAs and cytokines with regard to e.g., age, sex, treatment etc., while the individual correlations will be for an unadjusted association. No change in the manuscript.

Round 2

Reviewer 1 Report

Dear Authors,

thank you for addressing my comments and suggestions, I find your manuscript interesting and my suggestion is to accept it in present form.

Reviewer 2 Report

No more comments on the revised paper.